# Robust Tracking Control of Wheeled Mobile Robot Based on Differential Flatness and Sliding Active Disturbance Rejection Control: Simulations and Experiments

**DOI:** 10.3390/s24092849

**Published:** 2024-04-29

**Authors:** Amine Abadi, Amani Ayeb, Moussa Labbadi, David Fofi, Toufik Bakir, Hassen Mekki

**Affiliations:** 1Laboratory ImViA EA 7535, University of Bourgogne, 21000 Dijon, France; david.fofi@u-bourgogne.fr (D.F.); toufik.bakir@u-bourgogne.fr (T.B.); 2National Institute of Applied Science and Technology, Physics and Instrumentation Department, Tunis 1080, Tunisia; amani.ayeb@insat.ucar.tn; 3LIS UMR CNRS 7020, Aix-Marseille University, 13013 Marseille, France; moussa.labbadi@lis-lab.fr; 4NOCCS Laboratory, National School of Engineering of Sousse, University of Sousse, Sousse 4054, Tunisia; mekki.hassen@gmail.com

**Keywords:** differential flatness, sliding mode control, active disturbance rejection control, extended state observer, wheeled mobile robot

## Abstract

This paper proposes a robust tracking control method for wheeled mobile robot (WMR) against uncertainties, including wind disturbances and slipping. Through the application of the differential flatness methodology, the under-actuated WMR model is transformed into a linear canonical form, simplifying the design of a stabilizing feedback controller. To handle uncertainties from wheel slip and wind disturbances, the proposed feedback controller uses sliding mode control (SMC). However, increased uncertainties lead to chattering in the SMC approach due to higher control inputs. To mitigate this, a boundary layer around the switching surface is introduced, implementing a continuous control law to reduce chattering. Although increasing the boundary layer thickness reduces chattering, it may compromise the robustness achieved by SMC. To address this challenge, an active disturbance rejection control (ADRC) is integrated with boundary layer sliding mode control. ADRC estimates lumped uncertainties via an extended state observer and eliminates them within the feedback loop. This combined feedback control method aims to achieve practical control and robust tracking performance. Stability properties of the closed-loop system are established using the Lyapunov theory. Finally, simulations and experimental results are conducted to compare and evaluate the efficiency of the proposed robust tracking controller against other existing control methods.

## 1. Introduction

The domain of robotics finds mobile robots to be particularly intriguing, attracting considerable fascination and study. Designed to operate in dynamic settings, be it indoors or outdoors, these robots demonstrate the capacity to navigate autonomously or with minimal human input. Central to their functionality is their mobility, achieved through diverse locomotion methods, such as wheels, tracks, or legs. This mobility empowers them to traverse diverse terrains, overcoming obstacles encountered during their journeys. Recently, mobile robots have been used in various domains, including civilian, industrial, and military, to carry out diverse tasks such as surveillance [1], transportation [2], agricultural operations [3], and exploration [4]. Given the broad application spectrum and critical nature of tasks involving mobile robots, there exists an urgent need to develop performance tracking controllers to execute proposed missions with exceptional accuracy. However, achieving this objective remains a significant challenge due to the inherent under-actuation and nonlinearity in WMRs, constrained by nonholonomic limitations. Consequently, researchers have directed their efforts towards investigating the control of mobile robotic systems.

In the past few decades, substantial progress has been made in the field of tracking control for wheeled mobile robots (WMR) through the application of nonlinear control theory [5,6,7]. Among these control methodologies, linearization controllers, such as the flatness controller [8], have risen as a popular approach that can significantly simplify the controller design process. The flatness property is a technique used to define the dynamic behavior of nonlinear underactuated models by identifying a set of core system variables known as flat outputs. This perspective has significant implications for control systems, as will be demonstrated. The first step in flatness control involves generating a desired realizable trajectory that implicitly incorporates the system model. Following that, the nonlinear WMR model can be linearized, resulting in the canonical Brunovsky form [9,10]. This special form simplifies the concept of a feedback controller capable of achieving exact trajectory tracking. In fact, controlling a linear system is easier than controlling an underactuated nonlinear system, and this feature has encouraged researchers to use the properties of flatness in several application domains, such as the control of hydraulic systems [11], exoskeleton robots [12], microgrid [13], underwater robot [14], and quadrotor [15,16].

Numerous research studies on WMR have utilized the concept of flatness control. Abadi [17] introduced an approach for optimal path planning for WMR using the collocation method, flatness control, and spline curves. This method effectively reduces the time needed to compute optimal robot trajectories during navigation, which is crucial for real-world applications. Kaniche [18] proposed a flatness visual servoing control for WMR subjected to disturbances. Salah [19] developed an approach to generate the upper coverage trajectory of a mobile robot by leveraging flatness. Yakovlev [20] combined flatness control with predictive control to enable safe navigation of a WMR among static and dynamic obstacles.

There is always is a difference between the mathematical model describing the movement of WMR and reality. This difference is due to environmental phenomena neglected during modeling, such as wind, slipping, etc. The question that arises is how flatness control applied to WMR can ensure the accurate tracking of a desired trajectory despite the presence of uncertainties. To resolve this problem, a robust feedback controller must be combined with flatness, taking into account the impact of uncertainties to the model. Up to the present, there have been limited methods in the literature concerning the robustness issues of flatness systems. Among these approaches, the sliding mode control (SMC) has been successfully utilized in a variety of systems [21,22,23].

SMC is a robust control technique used to manage dynamic systems in the presence of uncertainties and disturbances. At its core, SMC aims to drive the system state onto a designated sliding surface within the state space. Once on this surface, the system’s behavior is constrained, allowing for effective regulation. SMC achieves this through discontinuous control actions, known as switching control, which dynamically alternate between different control laws. This switching mechanism ensures that the system remains on the sliding surface, enhancing robustness against external influences. Despite its effectiveness, SMC is associated with a phenomenon called chattering [24], characterized by rapid switching between control actions near the sliding surface. While chattering can theoretically improve tracking accuracy, it can lead to practical issues such as mechanical wear and high-frequency oscillations. To resolve this problem, numerous approaches have been suggested in the existing literature, such as high-order SMC [25], boundary layer [26], and active adaptive continuous nonsingular terminal sliding mode algorithm [27]. A frequently utilized approach for mitigating the chattering phenomenon involves incorporating the boundary layer technique within SMC. This entails replacing the sign function with a smooth function. However, this strategy presents its own set of challenges. Firstly, there exists a trade-off between the size of the boundary layer and the performance of SMC, which impacts the effectiveness of chattering reduction. Secondly, the robustness and accuracy of the system may not always be guaranteed within the boundary layer. Additionally, beyond addressing the chattering issue, achieving precise control of a robotic system necessitates knowledge about its state, typically obtained through real instruments, which can incur high costs and complicate the system’s structure. Moreover, in many instances, directly measuring certain system parameters may be impractical. To overcome these limitations, one potential solution involves implementing software sensors or observers, commonly referred to as virtual sensors. Therefore, to tackle both the reduced robustness resulting from the boundary layer approach and the challenge of state estimation, we propose a novel robust feedback controller that integrates the boundary layer sliding method with a disturbance observer.

In recent times, disturbance observers have emerged as potent tools for handling consolidated uncertainties, closely tied to disturbance-observer-based control. Within the domain of nonlinear disturbance observers, two notable approaches stand out: the uncertainty and disturbance estimator (UDE) [28] and the active disturbance rejection control [29] based on the extended state observer (ESO). In the UDE, only the disturbance is estimated, though in general, the observer equations depend on system states and inputs. Thus, a state observer is necessary unless all states are measurable. The idea of the ESO is to extend the original state vector by the disturbance vector and possibly, some of its time-derivatives, and then design a state observer for the extended system. ESO distinguishes itself by incorporating a dynamic model of disturbances or uncertainties into its estimation methodology, enabling it to identify and mitigate uncertainties not explicitly accounted for in the system model. The design of ESO is distinguished by its minimal dependence on system data and its freedom from the traditional system model, which simplifies its implementation process. Furthermore, several other types of disturbance observers are available, such as the nonlinear extended state observer (NLESO) [30], the adaptive extended state observer (AESO) [31], and the extended high-gain observer (EHGO) [32]. EHGO, part of the ESO family, stands out in two key aspects: it does not necessitate slow variations in disturbances, and it estimates a matched disturbance term originating from model uncertainty and external disturbances. Given the advantages offered by ESO, considerable research effort has been devoted to developing advanced controls for robotic systems.

In Ref. [33], Xie introduced a controller that integrates the backstepping technique with ESO to improve tracking performance for underwater robots. Additionally, Qi [34] improved the bandwidth of ESO to achieve more accurate disturbance estimation. Subsequently, they utilized a simple feedback controller to ensure attitude stabilization over a 3D hovering quadrotor system. In the work by Aole [35], an improved ADRC methodology for controlling lower limb exoskeletons is presented. The proposed approach integrates Linear ESO with a tracking differentiator, nonlinear state error feedback, and a proportional controller. Simulation results demonstrate the effectiveness of the suggested ADRC in efficiently regulating the hip and knee movements of the robot in the presence of disturbances. Hu [36] integrated a predictive control technique with ESO for unmanned underwater vehicles, offering a solution to concurrently handle external disturbances and system measurement noises. Based on this observation, the main contributions of our research can be summarized as follows:The kinematic model for WMR is structured in a standard format that systematically tackles underactuation and transforms nonmatching disturbances into matching ones through a flatness-based approach;The designated trajectory is feasible in practice because of the concept of differential flatness, which equates differential flatness with controllability, ensuring its physical achievability;Continuous sliding mode control (SMC) is employed to eliminate chattering, an essential necessity for the efficient application of control in real-world scenarios;SMC is integrated with ESO for the uncertain kinematic WMR model. This strategy seeks to improve the practicality and resilience of the tracking controller by reducing chattering through boundary layer SMC and estimating the lumped disturbance affecting the WMRs via ESO, which is then employed as a feedforward compensation;The proposed control method was compared with several other control methods, including traditional flatness control, backstepping tracking control flatness-based sliding control, and flatness active disturbance rejection control and backstepping sliding active disturbance rejection control. These comparisons were validated through simulations conducted in Matlab/Simulink and experiments carried out on the TurtleBot WMR.

The structure of the remaining sections of this article is outlined as follows. Section 2 provides a thorough overview of the flatness control technique for WMR. Section 3 elaborates on the concept of flatness-based sliding mode tracking control of WMRs. The proposed robust tracking controller is delineated in Section 4. Section 5 and Section 6 present and discuss the results of simulations and experiments. Finally, Section 7 concludes the paper by summarizing the key findings and suggesting potential future directions.

## 2. Flatness-Based Tracking Control

In our study, we analyzed a differential two-wheeled mobile robot (see Figure 1) that consists of two independent active wheels and a third passive wheel (a standard freewheel). This robotic system is widely regarded as an effective trade-off between control ease and the degrees of freedom that enable the robot to meet mobility requirements. The configuration of the mobile robot with wheels can be described by the vector $q_{r}=\left[x,y,\theta \right]$. In this notation, *x* and *y* represent the coordinates of the robot’s center position in the stationary frame (O,X,Y), while θ represents the orientation angle of the robot. The state equation of the WMR kinematic model, neglecting uncertainties, is represented as follows:(1)$$\begin{array}{c} \dot{x}=cos\left(\theta \right)v\\ \dot{y}=sin\left(\theta \right)v\\ \dot{\theta }=w \end{array}$$

The robot’s translational and rotational velocities are denoted by *v* and *w*, respectively. The angular velocities of the right and left wheels ($w_{r}$ and $w_{l}$) can be defined as functions of the robot’s translational and rotational velocities as follows:(2)$$v=(\frac{w_{r}+w_{l}}{2})r$$
(3)$$w=(\frac{w_{r}-w_{l}}{2b})r$$
where the variables *r* and 2b represent the radius and distance between the wheels, respectively. The nonholonomic limitation is defined as follows, based on the nonslip requirement:(4)$$\dot{x}sin\theta -\dot{y}cos\theta =0$$

The accuracy of the tracking will be guaranteed through the flatness property, which involves describing all system states and inputs, as well as their finite time derivatives, within the framework of a flat output. Considering the following nonlinear system:(5)$$\begin{array}{c} \dot{x}=f\left(x,u\right) \end{array}$$
where $x\in R^{n}$ and $u\in R^{m}$ represent the state and the input vector.

The nonlinear system (Equation 5) is differentially flat if there exists an output λ in the following form:(6)$$\lambda =\xi \left(x,u,\dot{u},\cdots ,u^{\left(c\right)}\right)\in R^{m}$$
such that the state and the input can be expressed as follows: (7)$$\begin{array}{c} x=\kappa_{1}\left(\lambda ,\dot{\lambda },\ddot{\lambda }\cdots ,\lambda^{\left(a\right)}\right)\; \end{array}$$(8)$$\begin{array}{c} u=\kappa_{2}\left(\lambda ,\dot{\lambda },\ddot{\lambda }\cdots ,\lambda^{\left(a+1\right)}\right) \end{array}$$
where *a* and *c* are finite multi-indices, and ξ, $\kappa_{1}$, and $\kappa_{2}$ are smooth vector functions of the output vector λ and its derivatives. By introducing the functions $\kappa_{1}$ and $\kappa_{2}$, this flat output is composed of a set of variables that enable the parameterization of all other system variables: the state, the command, and also the output λ. Indeed, if the output of the system is defined by a relation of the form $\lambda =\Xi (x,u,\dot{u},\ldots ,u^{\left(p\right)})$, then necessarily, the quantities described in Equations (Equation 7) and (Equation 8) make it possible to affirm that there exists an integer *c* such that:(9)$$f=\Xi \left(\lambda ,\dot{\lambda },\ddot{\lambda }\cdots ,\lambda^{\left(c\right)}\right)$$

The flat output combines all unconstrained variables of the system since the components of λ are differentially independent. Alternatively, based on Equation (Equation 9), we can argue that the flat output λ solely relies on the state and the command. This would make it an endogenous variable of the system, in contrast to the state of an observer, which would be an example of an exogenous variable of the observed system. In addition, Lie–Bäcklund’s notion of differential equivalence [8] shows that the number of components of λ is the same as the number of components of the control:(10)$$\begin{array}{c} dim\lambda =dimu \end{array}$$

This fundamental characteristic allows us to determine the requisite number of independent variables needed in a model to establish its flatness. A key benefit of the flatness property lies in its facilitation of various transformations, such as diffeomorphism and feedback linearization. These transformations enable the conversion of a nonlinear system into a controllable linear system, where the flat outputs represent the state vector.

Several studies in the literature, including Ref. [37], have shown that the WMR kinematic modeling can be defined as a differentially flat model, where the positional coordinates denoted as $\lambda ={\left[\lambda_{11},\lambda_{21}\right]}^{T}={\left[x,y\right]}^{T}$ serve as the flat outputs. Therefore, the entire set of state and control components pertaining to the WMR system are expressed using the flat variable λ and its derivatives, as demonstrated below:(11)$$\theta =arctan\frac{{\dot{\lambda }}_{21}}{{\dot{\lambda }}_{11}}\;$$
(12)$$v=\sqrt{{\dot{\lambda }}_{11}^{2}+{\dot{\lambda }}_{21}^{2}}$$
(13)$$w=\frac{{\dot{\lambda }}_{11}{\ddot{\lambda }}_{21}-{\ddot{\lambda }}_{11}{\dot{\lambda }}_{21}}{{\dot{\lambda }}_{11}^{2}+{\dot{\lambda }}_{21}^{2}}$$

The differentially flat nature of the WMR’s kinematic model has been demonstrated in the literature by various researchers [37]. This implies that all the states and controls of the kinematic WMR model can be expressed as functions of λ and its derivatives. However, the noninvertible relationship between the control input vectors *w* and *v* and the highest derivatives of the flat output limits the development of static feedback linearization for the nonlinear WMR. To address this constraint, we incorporate the control input *v* into the kinematic model defined by Equation (Equation 1) by treating it as an additional state. As a result, we obtain a revised system that can be defined as follows:(14)$$\begin{array}{c} \dot{x}=cos\left(\theta \right)v\\ \dot{y}=sin\left(\theta \right)v\\ \dot{v}=u_{r1}\\ \dot{\theta }=u_{r2} \end{array}$$

The state and control inputs of the modified system defined by Equation (Equation 14) are represented by $X_{r}={\left[x,y,v,\theta \right]}^{T}$ and $u_{r1}=\dot{v}$ and $u_{r2}=w$. In order to establish a bijective relationship between the inputs $u_{r1}$, $u_{r2}$, and higher-order derivatives of $\lambda_{11}=x$, $\lambda_{21}=y$, we apply successive differentiations to the flat outputs until at least one of the input variables appears in the resulting expressions, as illustrated below:(15)$$\left[\begin{array}{c} {\ddot{\lambda }}_{11}\\ {\ddot{\lambda }}_{21} \end{array}\right]=B_{rob}\left[\begin{array}{c} u_{r1}\\ u_{r2} \end{array}\right]$$
where $B_{rob}$ is described as follows:(16)$$B_{rob}=\left[\begin{array}{cc} cos(\theta ) & -vsin(\theta )\\ sin(\theta ) & vcos(\theta ) \end{array}\right]$$

The matrix $B_{rob}$ is not singular if v≠0. In this case, we can define the control as follows:(17)$$\left[\begin{array}{c} u_{r1}\\ u_{r2} \end{array}\right]=B_{rob}^{-1}\left[\begin{array}{c} {\ddot{\lambda }}_{11}\\ {\ddot{\lambda }}_{21} \end{array}\right]$$

To arrive at the linearized system, referred to as the Burnovsky Form (BF), we can substitute the control input (Equation 17) into Equation (Equation 15). This substitution yields the following modified expression:(18)$$\left(BF_{1}\right)\left\{\begin{array}{c} {\dot{\lambda }}_{11}=\lambda_{12}\\ {\dot{\lambda }}_{12}=v_{1}\\ Y_{1}=\lambda_{11}=x \end{array}\right.\left(BF_{2}\right)\left\{\begin{array}{c} {\dot{\lambda }}_{21}=\lambda_{22}\\ {\dot{\lambda }}_{22}=v_{2}\\ Y_{2}=\lambda_{21}=y \end{array}\right.$$
where $v_{1}$ and $v_{2}$ represent a suitable feedback controller defined as follows:(19)$$\begin{array}{c} {\;v}_{1}={\ddot{\lambda }}_{xd}-\sigma_{x2}\left(\lambda_{12}-{\dot{\lambda }}_{xd}\right)-\sigma_{x1}\left(\lambda_{11}-\lambda_{xd}\right) \end{array}$$
(20)$$\begin{array}{c} {\;v}_{2}={\ddot{\lambda }}_{yd}-\sigma_{y2}\left(\lambda_{22}-{\dot{\lambda }}_{yd}\right)-\sigma_{y1}\left(\lambda_{21}-\lambda_{yd}\right) \end{array}$$
where $\lambda_{xd}$ and $\lambda_{yd}$ denote the desired trajectories for the flat output $\lambda_{11}$ and $\lambda_{21}$, respectively. Meanwhile, the controller gains are represented by $\sigma_{x1}$, $\sigma_{x2}$, $\sigma_{y1}$, and $\sigma_{y2}$. The polynomial of the Burnovsky system (Equation 18) can be defined as follows:(21)$$s^{2}+\sigma_{x2}s+\sigma_{x1}=s^{2}+2m_{x}\epsilon_{xc}+\epsilon_{xc}^{2}$$
(22)$$s^{2}+\sigma_{y2}s+\sigma_{y1}=s^{2}+2m_{y}\epsilon_{yc}+\epsilon_{yc}^{2}$$
where the parameters $m_{x}$ and $m_{y}$ are the damping coefficients, and $\epsilon_{xc}$ and $\epsilon_{yc}$ are the frequencies in Equations (Equation 21) and (Equation 22). We can calculate the controller gain as follows:(23)$$\begin{array}{c} \sigma_{x1}=\epsilon_{xc}^{2},\sigma_{x2}=2m_{x}\epsilon_{xc},\sigma_{y1}=\epsilon_{yc}^{2},\sigma_{y2}=2m_{y}\epsilon_{yc} \end{array}$$

By integrating the feedback law, as described in Equations (Equation 19) and (Equation 20), into the system (Equation 17), we can express the flatness-based tracking control (FBTC) utilized for the mobile robot in the following manner:(24)$$\begin{array}{c} \left[\begin{array}{c} u_{FBTCx}\\ u_{FBTCy} \end{array}\right]=B_{rob}^{-1}\left[\begin{array}{c} {\ddot{\lambda }}_{xd}-\sigma_{x2}{\dot{e}}_{1}-\sigma_{x1}e_{1}\\ {\ddot{\lambda }}_{yd}-\sigma_{y2}{\dot{e}}_{2}-\sigma_{y1}e_{2} \end{array}\right] \end{array}$$
where $e_{1}=\lambda_{11}-\lambda_{xd}$ and $e_{2}=\lambda_{21}-\lambda_{yd}$.

In ideal conditions where uncertainties such as wind and wheel slip are negligible in the kinematic model of the WMR, the control input defined by Equation (Equation 24) can achieve satisfactory tracking performance for the desired trajectory. However, it is practically impossible to have a model that accurately represents the real-world movement of the robot in all environmental conditions. As a result, the following section will focus on developing a robust tracking control for a WMR kinematic model that is subject to uncertainties.

## 3. Flatness-Based Sliding Tracking Control

In order to account for real-world conditions, we consider uncertainties such as slippage and external environmental disturbances when describing the kinematic model of WMR (Figure 2). As a result, the model is defined differently, as shown below.
(25)$$\left(Uncertain\;Kinematic\;Model\right)\left\{\begin{array}{c} \dot{x}=cos\left(\theta \right)v+v_{t}cos\left(\theta \right)+v_{s}sin\left(\theta \right)+p_{x}\\ \dot{y}=sin\left(\theta \right)v+v_{t}sin\left(\theta \right)-v_{s}cos\left(\theta \right)+p_{y}\\ \dot{\theta }=w+w_{s} \end{array}\right.$$
where the variables $p_{x}$ and $p_{y}$ represent the external environmental disturbances, indicating the potential influences from the surrounding conditions. On the other hand, $v_{t}$ and $v_{s}$ represent the slip velocities, where $v_{t}$ denotes the slip velocity along the forward direction and $v_{s}$ represents the slip velocity normal to it. Additionally, $w_{s}$ denotes the angular slip velocity. According to [37], it is assumed that the slippage phenomenon can be defined and bounded as follows:(26)$$\begin{array}{c} v_{t}\left(t\right)=v_{s}\left(t\right)=w_{s}\left(t\right)=\kappa_{1}v\left(t\right) \end{array}$$
(27)$$\left|\right|v_{t}\left|\right|\leq \varepsilon_{1}\left||v||,\;|\right|v_{s}\left|\right|\leq \varepsilon_{2}\left||v||,|\right|w_{s}\left|\right|\leq \varepsilon_{3}$$
where $\kappa_{1}$, $\varepsilon_{1}$, $\varepsilon_{2}$ and $\varepsilon_{3}$ are positive constants.

Assuming that $\lambda_{xd}$ and $\lambda_{yd}$ are the reference trajectories for $\lambda_{11}$ and $\lambda_{21}$, respectively, we can define the error dynamics as $e_{i}=\lambda_{i1}-\lambda_{id}$ for i=1,2. To achieve convergence of the tracking error $e_{i}$ to zero in the presence of uncertainties, we employ a sliding mode control approach that relies on the principles of the flatness law. By incorporating this control strategy, we aim to ensure robust and accurate tracking performance even in the face of system uncertainties. The design of the sliding mode control involves two essential stages: the choice of the sliding surface and the development of the control law. These steps play a crucial role in establishing an effective and stable sliding mode control strategy. The selection of the sliding surface determines the desired system behavior and convergence properties, while the design of the control law focuses on generating control signals that guide the system towards the desired sliding surface and ensure its maintenance on that surface. In the context of the tracking example for the WMR, we make use of the sliding variable $\sigma_{r}={\left[s_{x},s_{y}\right]}^{T}$ to represent the tracking error. To define the sliding surface, we consider the desired tracking behavior and express it as follows, taking into account the specific requirements of the system:(28)$$s_{x}={\dot{e}}_{1}+\beta_{1}e_{1}$$
(29)$$s_{y}={\dot{e}}_{2}+\beta_{2}e_{2}$$
where the gains $\beta_{1}$ and $\beta_{2}$ can be selected using pole-placement techniques to ensure the asymptotic convergence of the tracking errors $e_{1}=\lambda_{11}-\lambda_{xd}$ and $e_{2}=\lambda_{21}-\lambda_{yd}$ to zero. In this tracking example, the sliding variable $\sigma_{r}={\left[s_{x},s_{y}\right]}^{T}$ is chosen as the tracking error. Therefore, the sliding surface for the WMR can be defined as follows:(30)$${\dot{e}}_{1}+\beta_{1}e_{1}=0$$
(31)$${\dot{e}}_{2}+\beta_{2}e_{2}=0$$

As suggested by Mauledoux [38], to guarantee that the sliding surface $\sigma_{r}=0$ is attractive, we can enforce the dynamics of $\sigma_{r}$ as follows:(32)$${\dot{\sigma }}_{r}=-k_{i}sgn\left(\sigma_{r}\right)$$
where the standard signum function is denoted by sgn, and $k_{i}$ (i=1,2) is a constant. One approach to proving the error dynamics stability is to analyze the following Lyapunov function:(33)$$V_{s}=\frac{1}{2}\sigma_{r}^{2}$$

The derivative of $V_{s}$ is defined as follows:(34)$${\dot{V}}_{s}=\sigma_{r}{\dot{\sigma }}_{r}$$

We can conclude that $V_{s}$ is a positive function and its derivative ${\dot{V}}_{s}$ is negative or zero. Hence, the system exhibits asymptotic Lyapunov stability. Using Equations (Equation 28), (Equation 29) and (Equation 32) we obtain:(35)$$-k_{1}sign\left(s_{x}\right)={\ddot{e}}_{1}+\beta_{1}{\dot{e}}_{1}$$
(36)$$-k_{2}sign\left(s_{y}\right)={\ddot{e}}_{2}+\beta_{2}{\dot{e}}_{2}$$

As a result, by using Equations (Equation 35) and (Equation 36), we can obtain:(37)$${\ddot{\lambda }}_{11}={\ddot{\lambda }}_{xd}-\beta_{1}{\dot{e}}_{1}-k_{1}sgn\left(s_{x}\right)$$
(38)$${\ddot{\lambda }}_{21}={\ddot{\lambda }}_{yd}-\beta_{2}{\dot{e}}_{2}-k_{2}sgn\left(s_{y}\right)$$

Substituting ${\ddot{\lambda }}_{11}$ and ${\ddot{\lambda }}_{21}$ with their new expressions defined by Equations (Equation 35) and (Equation 36) in the control defined by (Equation 17), the flatness-based sliding mode tracking controller (FSMC) applied to WMR is defined as follows:(39)$$\left[\begin{array}{c} u_{FSMCx}\\ u_{FSMCy} \end{array}\right]=B_{rob}^{-1}\left[\begin{array}{c} {\ddot{\lambda }}_{xd}-\beta_{1}{\dot{e}}_{1}-k_{1}sgn\left(s_{x}\right)\\ {\ddot{\lambda }}_{yd}-\beta_{2}{\dot{e}}_{2}-k_{2}sgn\left(s_{y}\right) \end{array}\right]$$

The FSMC defined by Equation (Equation 39) contains a discontinuous control term due to the function sgn(σ). Although selecting sufficiently large values for $k_{1}$ and $k_{2}$ can achieve convergence to sliding variable in limited time and provide robustness against perturbations, it also causes the phenomenon of chattering. Thus, to avoid this problem, the function sgn(σ) can be replaced by the function Sat defined as follows:(40)$$Sat\left(\sigma_{r}\right)\left\{\begin{array}{c} \frac{\sigma_{r}}{a_{s}}\;if\;\left|\sigma_{r}\right|\leq a_{s}\\ sgn\left(\sigma_{r}\right)\;if\;\left|\sigma_{r}\right|>a_{s} \end{array}\right.$$
where $a_{s}$ is the width of the threshold of the saturation function.

The thickness of the boundary layer, denoted as $a_{s}$, within the saturation function stands as a pivotal parameter influencing the efficacy of the sliding mode controller. As the value of $a_{s}$ increases, the approximation diverges more from the ideal sgn function, resulting in enhanced reduction of chattering. However, this improvement comes at the cost of diminished robustness. Conversely, if the value of the parameter $a_{s}$ is reduced, the change of the control signal will be too frequent, which leads to inevitable chatter of the control signal. Therefore, a variable-thickness boundary layer $a_{s}$ is tailored to strike a balance between mitigating chattering and upholding system robustness amid uncertainties. In the upcoming section, the FSMC described by Equation (Equation 39) will be integrated with active disturbance rejection control to enhance the robustness lost by the Sat function and maintain the advantage of reducing chattering.

## 4. Proposed Robust Tracking Controller

In this section, we introduce a novel cascade control strategy that utilizes a combination of flatness property, active disturbance rejection control (ADRC), and boundary layer sliding mode control to solve the problem of reduced robustness obtained when replacing the function sgn by the function sat in the FSMC defined by Equation (Equation 39). Given the uncertain kinematic model (Equation 25), we can obtain the following relationship by differentiating $\lambda_{11}$ and $\lambda_{21}$ until the input terms $u_{1}$ and $u_{2}$ become evident:(41)$$\left[\begin{array}{c} {\ddot{\lambda }}_{11}\\ {\ddot{\lambda }}_{21} \end{array}\right]=B_{rob}\left[\begin{array}{c} u_{r1}\\ u_{r2} \end{array}\right]+C_{rob}+D_{rob}\left[\begin{array}{c} u_{r1}\\ u_{r2} \end{array}\right]$$
where $C_{rob}$ and $D_{rob}$ are defined as follows:(42)$$C_{rob}=\left[\begin{array}{c} cos\left(\theta \right)\left(v_{s}w_{s}+{\dot{v}}_{t}\right)+sin\left(\theta \right)\left({\dot{v}}_{s}-vw_{s}-v_{t}w_{s}\right)+{\dot{p}}_{x}\\ sin\left(\theta \right)\left(v_{s}w_{s}+{\dot{v}}_{t}\right)-cos\left(\theta \right)\left({\dot{v}}_{s}-vw_{s}-v_{t}w_{s}\right)+{\dot{p}}_{y} \end{array}\right],D_{rob}=\left[\begin{array}{cc} 0 & -v_{t}sin\left(\theta \right)+v_{s}cos\left(\theta \right)\\ 0 & v_{t}cos\left(\theta \right)+v_{s}sin\left(\theta \right) \end{array}\right]$$

By utilizing the control input described in Equation (Equation 17) on system (Equation 41), we achieve:(43)$$\ddot{\lambda }=v+\delta$$
where $\ddot{\lambda }={\left[{\ddot{\lambda }}_{11},{\ddot{\lambda }}_{21}\right]}^{T}$,$v={\left[v_{1},v_{2}\right]}^{T}$ and $\delta ={\left[\delta_{1},\delta_{2}\right]}^{T}=d_{rob}B_{rob}^{-1}v+C_{rob}$.

Rewriting Equation (Equation 43) in terms of two linear integrator systems subject to perturbation yields the following expressions:(44)$$MBF_{1}\left\{\begin{array}{c} {\dot{\lambda }}_{11}=\lambda_{12}\\ {\dot{\lambda }}_{12}=v_{1}+\delta_{1}\\ Y_{1}=\lambda_{11} \end{array}\right.\;MBF_{2}\left\{\begin{array}{c} {\dot{\lambda }}_{21}=\lambda_{22}\\ {\dot{\lambda }}_{22}=v_{2}+\delta_{2}\\ Y_{2}=\lambda_{21} \end{array}\right.$$

Consider $\Delta_{1}$ and $\Delta_{2}$ as the differentials of $\delta_{1}$ and $\delta_{2}$ with respect to time *t*, respectively. We assume that both $\delta_{i}$ and $\Delta_{i}$ (i=1,2) are bounded. In practical applications, determining the actual values of the lumped disturbances $\delta_{1}$ and $\delta_{2}$ that affect the system is considered a challenging problem. Hence, an observer is required to estimate these values.

### 4.1. ESO Design

The extended state observer (ESO) plays a vital role in system control by simultaneously estimating the system states and uncertainties. This capability enables the ESO to effectively reject or compensate for disturbances, enhancing the system’s robustness and performance. The ESO takes into account all factors that affect the system and treats parameter uncertainties and external perturbations as a single observed disturbance. The ESO is named as such because it estimates uncertainties as an extended state. Its benefits include not being reliant on the mathematical model of the system, as well as having a straightforward implementation and demonstrating good performance. Consider $\lambda_{13}=\delta_{1}$, $\alpha_{23}=\delta_{2}$ as an extended state for system (Equation 44). The latter can be expressed as follows:(45)$$\begin{array}{c} \left\{\begin{array}{c} {\dot{\lambda }}_{11}=\lambda_{12}\\ {\dot{\lambda }}_{12}=\lambda_{13}+v_{1}\\ {\dot{\lambda }}_{13}=\Delta_{1}\\ Y_{1}=\lambda_{11} \end{array}\right.\left\{\begin{array}{c} {\dot{\lambda }}_{21}=\lambda_{22}\\ {\dot{\lambda }}_{22}=\lambda_{23}+v_{2}\\ {\dot{\lambda }}_{23}=\Delta_{2}\\ Y_{1}=\lambda_{21} \end{array}\right. \end{array}$$

We can express systems (Equation 45) in matrix form as follows:(46)$$\begin{array}{c} \left\{\begin{array}{c} {\dot{\lambda }}_{1}=A_{x}\lambda_{1}+B_{x}v_{1}+E_{x}\Delta_{1}\\ Y_{1}=C_{x}\lambda_{1} \end{array}\right. \end{array}$$
(47)$$\begin{array}{c} \left\{\begin{array}{c} {\dot{\lambda }}_{2}=A_{y}\lambda_{2}+B_{y}v_{2}+E_{y}\Delta_{2}\\ Y_{2}=C_{y}\lambda_{2} \end{array}\right. \end{array}$$
where $\lambda_{1}={\left[\lambda_{11},\lambda_{12},\lambda_{13}\right]}^{T}$, $\lambda_{2}={\left[\lambda_{21},\lambda_{22},\lambda_{23}\right]}^{T}$, $A_{x}=A_{y}=\left[\begin{array}{ccc} 0 & 1 & 0\\ 0 & 0 & 1\\ 0 & 0 & 0 \end{array}\right],B_{x}=B_{y}=\left[\begin{array}{c} 0\\ 1\\ 0 \end{array}\right],$ $C_{x}=C_{y}=\left[\begin{array}{ccc} 1 & 0 & 0 \end{array}\right],$ $E_{x}=E_{y}={\left[\begin{array}{ccc} 0 & 0 & 1 \end{array}\right]}^{T}$. The expression for the Extended State Observer (ESO) corresponding to each extended system (Equation 46) and (Equation 47) can be given as follows:(48)$${\dot{\hat{\lambda }}}_{1}=A_{x}{\hat{\lambda }}_{1}+B_{x}v_{x}+\alpha_{gx}C_{x}\left(\lambda_{1}-{\hat{\lambda }}_{1}\right)$$
(49)$${\dot{\hat{\lambda }}}_{2}=A_{y}\hat{\lambda }2+B_{y}v_{y}+\alpha_{gy}C_{y}\left(\lambda_{2}-{\hat{\lambda }}_{2}\right)$$
where $\alpha_{gx}={\left[\alpha_{11},\alpha_{12},\alpha_{13}\right]}^{T}$, $\alpha_{gy}={\left[\alpha_{21},\alpha_{22},\alpha_{23}\right]}^{T}$. To determine the observer gains $\alpha_{ij}$(i=1,2,3), (j=1,2,3), we can adopt the methodology proposed by Gao [39] outlined in the following manner:(50)$$s^{3}+\alpha_{11}s^{2}+\alpha_{12}s+\alpha_{13}={\left(s+\gamma_{xo}\right)}^{3}$$
(51)$$s^{3}+\alpha_{21}s^{2}+\alpha_{22}s+\alpha_{23}={\left(s+\gamma_{yo}\right)}^{3}$$

The choice of $\gamma_{xo}$ and $\gamma_{yo}$ is made to ensure that Equations (Equation 50) and (Equation 51) form Hurwitz polynomials with respect to the complex variable. The observer gain can be formulated as a function of the ESO bandwidth by utilizing Equations (Equation 50) and (Equation 51), as demonstrated below:(52)$$\begin{array}{c} \alpha_{11}=3\gamma_{xo},\alpha_{12}=3\gamma_{xo}^{2},\alpha_{13}=\gamma_{xo}^{3}\\ \alpha_{21}=3\gamma_{yo},\alpha_{22}=3\gamma_{yo}^{2},\alpha_{23}=\gamma_{yo}^{3}. \end{array}$$

The observer error associated with each ESO can be defined by employing Equations (Equation 46)–(Equation 49) as follows:(53)$${\dot{\hat{e}}}_{x}={\dot{\lambda }}_{1}-{\dot{\hat{\lambda }}}_{1}=\left(A_{x}-\alpha_{gx}C_{x}\right){\hat{e}}_{x}+E_{x}\Delta_{1}$$
(54)$${\dot{\hat{e}}}_{y}={\dot{\lambda }}_{2}-{\dot{\hat{\lambda }}}_{2}=\left(A_{y}-\alpha_{gy}C_{y}\right){\hat{e}}_{y}+E_{y}\Delta_{2}$$

It is possible to express Equations (Equation 53) and (Equation 54) in matrix form as shown below:(55)$$\dot{\hat{e}}=\hat{H}\hat{e}+E_{d}$$
where $\hat{e}={\left[{\hat{e}}_{x},{\dot{\hat{e}}}_{x},{\ddot{\hat{e}}}_{x},{\hat{e}}_{y},{\dot{\hat{e}}}_{y},{\ddot{\hat{e}}}_{y}\right]}^{T}$, H^=H^10303H2^, ${\hat{H}}_{1}=\left[\begin{array}{ccc} -\alpha_{11} & 1 & 0\\ -\alpha_{12} & 0 & 1\\ -\alpha_{13} & 0 & 0 \end{array}\right]$, ${\hat{H}}_{2}=\left[\begin{array}{ccc} -\alpha_{21} & 1 & 0\\ -\alpha_{22} & 0 & 1\\ -\alpha_{23} & 0 & 0 \end{array}\right]$, $E_{d}={\left[\begin{array}{cccccc} 0 & 0 & \Delta_{1} & 0 & 0 & \Delta_{2} \end{array}\right]}^{T}$.

 **Lemma** **1.** 
*In Equation (Equation 55), the boundedness of $\lim_{t\to \infty }\hat{e}\left(t\right)$ can be guaranteed if at least one of the following two conditions is satisfied:*

*$\delta_{i}<n_{1}$, i=1,2 for all time t;*

*$\Delta_{i}<n_{2}$, i=1,2 for all time t.*


*Asymptotic stability of the estimated error dynamics can be achieved when the values of $\delta_{i}$, i=1,2, are either directly obtained or assumed to be constant, leading to $\Delta_{i}=0$, i=1,2. In this scenario, the positive constants $n_{1}$ and $n_{2}$ play a vital role in ensuring the system’s stability. Lemma 1, as stated in Zhang et al. [40], establishes that the roots of the matrix $\hat{H}$ in Equation (Equation 55) reside in the left half plane. This result is ensured by the nonnegativity of the bandwidths $\gamma_{xo}$ and $\gamma_{yo}$. Consequently, it can be deduced that the estimated error dynamics described by Equations (Equation 53) and (Equation 54) are asymptotically stable.*


### 4.2. New Robust Feedback Controller

The feedback controller presented in Equations (Equation 19) and (Equation 20) relies on state measurements, but except for $\lambda_{11}$ and $\lambda_{21}$, the remaining states cannot be accurately measured. To solve this problem, the state estimation obtained through the two ESOs defined in Equations (Equation 48) and (Equation 49) are used instead. Furthermore, in order to simplify the compensation of the lumped disturbances $\delta_{1}$ and $\delta_{2}$, they are replaced by their approximations, ${\hat{\delta }}_{1}$ and ${\hat{\delta }}_{2}$. By incorporating the results of the extended state observers (ESOs), a robust feedback controller can be developed in the following manner:(56)$${\;v}_{SADRCx}={\ddot{\lambda }}_{xd}-\beta_{1}{\dot{\hat{e}}}_{1}-k_{1}sat\left({\hat{s}}_{x}\right)-{\hat{\delta }}_{1}$$
(57)$${\;v}_{SADRCy}={\ddot{\lambda }}_{yd}-\beta_{2}{\dot{\hat{e}}}_{2}-k_{2}sat\left({\hat{s}}_{y}\right)-{\hat{\delta }}_{2}$$
according to the sliding mode active disturbance rejection control feedback given in Equations (Equation 56) and (Equation 57), we can obtain the new robust tracking controller named Flatness-Sliding-Active-Disturbance-Rejection Control (FSADRC), defined as follows:(58)$$\begin{array}{c} \left[\begin{array}{c} u_{FSADRCx}\\ u_{FSADRCy} \end{array}\right]=B_{r}^{-1}\left[\begin{array}{c} {\ddot{\lambda }}_{xd}-\beta_{1}{\dot{\hat{e}}}_{1}-k_{1}sat\left({\hat{s}}_{x}\right)-{\hat{\delta }}_{1}\\ {\ddot{\lambda }}_{yd}-\beta_{2}{\dot{\hat{e}}}_{2}-k_{2}sat\left({\hat{s}}_{y}\right)-{\hat{\delta }}_{2} \end{array}\right] \end{array}$$
where ${\hat{e}}_{r1}={\hat{\lambda }}_{11}-\lambda_{xd}$ and ${\hat{e}}_{r2}={\hat{\lambda }}_{21}-\lambda_{yd}$. The schematic diagram presented in Figure 3 illustrates the principle of trajectory tracking control for a mobile robot.

### 4.3. Stability Analysis of the Closed-Loop System

This section will address the stability analysis of the tracking error systems for *x* and *y*, utilizing the estimation error defined by Equations (Equation 53) and (Equation 54). In order to prove the stability of the error dynamics of position *x*, Lyapunov’s function is chosen as follows:(59)$$V_{sx}=\frac{1}{2}s_{x}^{2}$$
where $s_{x}={\dot{e}}_{1}+\beta_{1}e_{1}={\dot{\lambda }}_{11}-{\dot{\lambda }}_{xd}+\beta_{1}\left(\lambda_{11}-\lambda_{xd}\right),\lambda_{11}=x,\lambda_{xd}=x_{d}$.

We can define the derivative of the Lyapunov function $V_{sx}$ as follows:(60)$${\dot{V}}_{sx}=s_{x}{\dot{s}}_{x}=s_{x}\left({\ddot{\lambda }}_{11}-{\ddot{\lambda }}_{xd}+\beta_{1}\left({\dot{\lambda }}_{11}-{\dot{\lambda }}_{xd}\right)\right)$$

When replacing ${\ddot{\lambda }}_{11}$ by its Equation (Equation 44) defined by ${\dot{\lambda }}_{11}=v_{1}+\delta_{1}$, we obtain:(61)$${\dot{V}}_{sx}=s_{x}{\dot{s}}_{x}=s_{x}\left(v_{1}+\delta_{1}-{\ddot{\lambda }}_{xd}+\beta_{1}\left({\dot{\lambda }}_{11}-{\dot{\lambda }}_{xd}\right)\right)$$

When $v_{1}$ represents the feedback controller, substituting it with the proposed robust feedback tracking control, denoted as $v_{SADRCx}$ defined by Equation (Equation 56), yields:(62)$$\begin{array}{c} \begin{array}{c} {\dot{V}}_{sx}=s_{x}{\dot{s}}_{x}=s_{x}\left({\ddot{\lambda }}_{xd}-\beta_{1}\left({\dot{\hat{\lambda }}}_{11}-{\dot{\lambda }}_{xd}\right)-k_{1}sat\left({\hat{s}}_{x}\right)-{\hat{\delta }}_{1}+\delta_{1}-{\ddot{\lambda }}_{xd}+\beta_{1}\left({\dot{\lambda }}_{11}-{\dot{\lambda }}_{xd}\right)\right)\\ {\dot{V}}_{sx}=s_{x}{\dot{s}}_{x}=s_{x}\left(\beta_{1}\left({\dot{\lambda }}_{11}-{\dot{\hat{\lambda }}}_{11}\right)+\delta_{1}-{\hat{\delta }}_{1}-k_{1}sat\left({\hat{s}}_{x}\right)\right) \end{array} \end{array}$$
where $s_{x}$ is defined as follows:(63)$$Sat\left(s_{x}\right)=\left\{\begin{array}{c} \frac{s_{x}}{a_{sx}}\;if\;\left|s_{x}\right|\leq a_{sx}\\ sgn\left(s_{x}\right)\;if\;\left|s_{x}\right|>a_{sx} \end{array}\right.$$

Concerning the stability and boundedness of the ESO defined by Equation (Equation 53), it can be achieved by choosing $\alpha_{gx}$ in such a way that the eigenvalues of $A_{x}-\alpha_{gx}C_{x}$ are negative, indicating poles in the left-hand plane, and ensuring that uncertainty is bounded. As a result, the error $\dot{\hat{e}}x\to 0$. This implies that $\hat{\lambda }11\to \lambda_{11}$, ${\hat{\delta }}_{1}\to \delta_{1}$, and ${\hat{s}}_{x}\to s_{x}$. In this scenario, the Lyapunov function defined by Equation (Equation 62) is formulated as follows:(64)$$\begin{array}{c} \begin{array}{c} {\dot{V}}_{sx}=-s_{x}\left(k_{1}sat\left(s_{x}\right)\right) \end{array} \end{array}$$

Since $Sat(s_{x})$, defined by Equation (Equation 63), is divided into two segments, the proof process will be analyzed in two cases. In the first scenario, when the saturation function is defined as described by:(65)$$Sat\left(s_{x}\right)=\frac{s_{x}}{a_{sx}}$$

Moreover, the Lyapunov function is defined as follows:(66)$$\begin{array}{c} \begin{array}{c} {\dot{V}}_{sx}=-\frac{k}{a_{sx}}\left(s_{x}^{2}\right)\leq 0 \end{array} \end{array}$$

Alternatively, when the saturation function is given by:(67)$$Sat\left(s_{x}\right)=sgn\left(s_{x}\right)$$
the Lyapunov function takes the form:(68)$$\begin{array}{c} \begin{array}{c} {\dot{V}}_{sx}=-k_{1}s_{x}sgn\left(s_{x}\right)\leq 0 \end{array} \end{array}$$

Thus, based on Equations (Equation 66) and (Equation 68), it can be concluded that the Lyapunov function ${\dot{V}}_{sx}$ is negative regardless of the definition of the function $Sat(s_{x})$. As a result, the tracking error of the position *x* is stable. Similarly, the same conclusions about the stability of the closed-loop system *y* can be drawn.

## 5. Simulation Results

This section presents simulation tests to validate the efficacy and superiority of the suggested controller, flatness sliding active disturbance rejection control (FSADRC), as defined by Equation (Equation 58). The proposed control is evaluated against flatness sliding mode control (FSMC), represented by Equation (Equation 39), and flatness-based tracking control (FBTC), as defined in Equation (Equation 24), using computer simulation results. The parameters of the WMR are r=0.1m, b=0.15m. To enhance the observation and comparison of the simulation results, we have chosen two types of reference trajectories: a circular path and a Bézier curve. Additionally, we also consider two different scenarios of perturbation. The controller design parameters of FBTC, FSMC, and FSADRC are chosen as $m_{x}=m_{y}=1$, $\epsilon_{xc}=\epsilon_{yc}=2$, $\beta_{1}=\beta_{2}=5$, and $k_{1}=k_{2}=10$. As suggested by Gao [39], it is advisable to select the observer bandwidth to be sufficiently higher than the controller bandwidth. This ensures that the observer dynamics remain faster than the system dynamics, enabling effective disturbance estimation and compensation. In our case, we have chosen observer bandwidths of $\gamma_{xo}=\gamma_{yo}=6\;\mathrm{rad}/s$ to fulfill this requirement and ensure robust performance of the control system. To ensure that the sliding mode control system achieves both satisfactory dynamic and steady-state performance, and to prevent chatter in the control signal, the cut and dry method is frequently employed to establish the thickness of the boundary layer. Specifically, in this case, $a_{sx}=a_{sy}=0.3$ is chosen.

### 5.1. First Scenario

In this simulation, we consider that slip velocities $v_{t}$ and $v_{s}$ can be up to 30% of the forward speed. Thus, $\kappa_{1}=0.3$. In addition, the WMR is subjected to constant wind perturbation defined as follows:(69)$$p_{x}=p_{y}=3\;m/s,w_{s}=0.5\;\mathrm{rad}/s$$

The reference trajectory considered in this scenario is a circle, which is defined by the following equation:(70)$$x_{r}=cos\left(t\right),y_{r}=sin\left(t\right)$$

The performance of the uncertain WMR systems under different control strategies, namely FBTC, FSMC, and FSADRC, is depicted in Figure 4. Figure 5 shows the results of the estimated lumped disturbance affecting the *x* and *y* channels obtained using the extended state observer (ESO). Figure 6 illustrates the control input applied to the wheeled mobile robot under the conditions of the first scenario. The simulation results indicate that the uncertainty caused by slow wind perturbation and slip decreases the tracking performance in trajectory following, rendering FBTC ineffective as a controller. On the other hand, both FSMC and FSADRC demonstrate robustness in handling the overall disturbance affecting the WMR model. These controllers exhibit the ability to mitigate disturbances and successfully maintain the desired trajectory of the WMR system. Consequently, it can be inferred that controllers that disregard uncertain models, despite being feedback controllers, may exhibit unsatisfactory performance. The fundamental distinction between the FSMC and FSADRC controllers lies in their design methodologies and approaches. FSMC relies on finely-tuned gains to achieve disturbance rejection, which can lead to chattering due to the relatively high gain values. In contrast, FSADRC combines the advantages of the boundary layer method to minimize chattering and an ESO to estimate and eliminate lumped disturbance.

### 5.2. Second Scenario

The objective of this simulation is to create and follow a trajectory for a robot, starting from an initial state where x(0)=y(0)=0, and reaching a final state specified by x(10)=3.5 and y(10)=5. This trajectory must navigate through a room containing obstacles, while also considering time-varying wind disturbances and slipping. The desired trajectory should meet the following criteria: minimizing energy consumption, maneuvering around static obstacles, and adhering to the specified state constraints as follows:(71)$$\begin{array}{c} 0\;m\leq \lambda_{xd}\leq \;4\;m,\;0\;m\leq \lambda_{yd}\leq \;6\;m \end{array}$$

The optimal trajectory generation method proposed in [17] offers a solution to obtain the desired trajectory by solving a nonlinear optimization problem. By integrating the principles of flatness, the collocation method, and B-spline functions, this method efficiently generates trajectories while guaranteeing constraint satisfaction. To ensure consistency in the simulation results, the parameters for all three controllers remain unchanged from the previous simulations. Considering an uncertain initial condition of $\hat{x}\left(0\right)=1$ and $\hat{y}\left(0\right)=1$ for the wheeled mobile robot (WMR), we further specify that the slip velocities $v_{t}$ and $v_{s}$ can potentially reach up to 50% to 70%. In addition, we take into account the influence of sinusoidal wind disturbances. In contrast to the initial scenario, the disturbance signals consist of combinations of multi-frequency sinusoidal signals representing time-varying disturbances, particularly wind, defined as follows:(72)$$p_{x}=p_{y}=1.5+2.5sin\left(4t\right)+4.5cos\left(2t\right),w_{s}=1.5+3cos\left(2t\right)$$

The simulation results regarding trajectory tracking performance of the second scenario are depicted in Figure 7. Based on these figures, it can be observed that the WMR system experiences significant divergence from the desired trajectory when affected by slippage and external disturbances, rendering FBTC ineffective as a controller. The FSMC controller’s intervention through the sliding mode’s discontinuous term eliminates uncertainty effects and maintains the stability of the closed-loop control. However, as shown in Figure 8, the presence of chattering in the FSMC control signals negatively impacts the system’s behavior. Hence, it can be inferred that while FSMC is a robust control approach, its practical applicability is quite restricted. Therefore, developing a control approach capable of mitigating the chattering effect while maintaining the robustness advantage provided by SMC is necessary. The results of the lumped disturbance estimation for this simulation are illustrated in Figure 9. According to the simulation findings, the mobile robot satisfactory trajectory tracking performance when confronted with model disturbances and uncertain initial conditions while employing the FSADRC controller. Of greater significance, the proposed control methodology achieves superior tracking of the desired trajectory, devoid of the chattering phenomenon, and enhances tracking performance against aggressive disturbances.

## 6. Tracking the Experimental Results of a Wheeled Mobile Robot

This section outlines experiments conducted with the TurtleBot3, a Wheeled Mobile Robot (WMR), to evaluate a proposed methodology. The TurtleBot provides a cost-effective platform for researchers to explore and validate control algorithms without requiring expensive robotic systems. Its compatibility with the Robot Operating System (ROS) enhances its functionalities, offering resources for algorithm development and experimentation. With LiDAR, IMU, and wheel encoders onboard, the TurtleBot3 provides precise environmental feedback, facilitating algorithm optimization. Researchers can augment the system with additional sensors or hardware components to evaluate various control algorithms across diverse scenarios. To facilitate the observation and comparison of experimental results, we have selected two types of reference trajectories: an eight-shaped path and a Bézier curve. Additionally, we have considered two different scenarios of perturbation: the first involves slowly time-varying disturbances, while the second entails aggressive time-varying disturbances. For further validation, the performance of the proposed control method is compared with other state-of-the-art control techniques such as backstepping tracking control (BTC) [41], flatness active disturbance rejection control (FADRC) introduced in [42], flatness-based tracking control (FBTC) as defined by Equation (Equation 39), and backstepping sliding active disturbance rejection control (BSADRC) [43]. The controller design parameters selected for the experimental results are identical to those chosen for the simulation results.

### 6.1. First Experiment with Slowly Time-Varying Disturbances

In this experiment, eight shapes were chosen for the reference trajectory, as outlined below:(73)$$x_{r}=2cos\left(t\right),y_{r}=-2sin\left(t\right)$$

To replicate real-world navigation conditions for the WMR, high-speed fans are utilized in the laboratory to simulate windy environments. Additionally, a stick is employed to disturb the castors of the WMR, creating slipping incidents, thus adding further realism to the testing environment. Figure 10 illustrates the real-time tracking of the eight-shaped reference trajectory of the WMR using the proposed control method described in this paper.

Simulation of the experiment under identical conditions reveal tracking trajectories in Figure 11. Figure 12 illustrates lumped disturbance estimation, while Figure 13 displays control torques. Based on the experimental results shown in Figure 11, it is evident that FADRC, FSADRC, and BSADRC methods excel at tracking trajectories even in the face of genuine uncertainty. Conversely, the FBTC and BTC methods demonstrate significant shortcomings when it comes to handling uncertainties. To assess the superiority of the proposed control, we will conduct a thorough study in the subsequent section. This study will include a quantitative analysis of the controllers under more severe disturbance conditions.

### 6.2. Second Experiment with Aggressive Time-Varying Disturbances

In this experiment, we intensify the frequency of disturbance variation generated by the industrial ventilator and subject the robot to aggressive impacts with a stick to assess the effectiveness of the proposed controller. Additionally, we adopt the eighth-order Bézier curve as a reference trajectory for both the *x* and *y* positions, defined as follows:(74)$$\begin{array}{c} \begin{array}{c} \lambda_{xd}=x_{r}=P_{x0}{\left(1-t\right)}^{8}+8P_{x1}{\left(1-t\right)}^{7}t+28P_{x2}{\left(1-t\right)}^{6}t^{2}P+56P_{x3}{\left(1-t\right)}^{5}t^{3}+\cdots \\ 70P_{x4}{\left(1-t\right)}^{4}t^{4}+56P_{x5}{\left(1-t\right)}^{3}t^{5}+28P_{x6}{\left(1-t\right)}^{2}t^{6}+8P_{x7}\left(1-t\right)t^{7}+P_{x8}t^{8}.\\ \lambda_{yd}=y_{r}=P_{y0}{\left(1-t\right)}^{8}+8P_{y1}{\left(1-t\right)}^{7}t+28P_{y2}{\left(1-t\right)}^{6}t^{2}P+56P_{y3}{\left(1-t\right)}^{5}t^{3}+\cdots \\ 70P_{y4}{\left(1-t\right)}^{4}t^{4}+56P_{y5}{\left(1-t\right)}^{3}t^{5}+28P_{y6}{\left(1-t\right)}^{2}t^{6}+8P_{y7}\left(1-t\right)t^{7}+P_{y8}t^{8}. \end{array} \end{array}$$
where $P_{xj}$, $P_{yj}$, and j=0⋯8 represent the control parameters of the reference trajectory. These parameters may vary depending on several factors, including the robot’s initial position, the desired final position, and constraints such as obstacle avoidance. As an example, we select control parameters that allow the WMR to transition from its initial state $q_{r}\left(0\right)={\left[0,0,0\right]}^{T}$ to the desired final state $q_{r}\left(20\right)={\left[2,2,0\right]}^{T}$. The tracking experiment results of the WMR under aggressive time-varying disturbances are depicted in Figure 14. In Figure 15, the lumped disturbance affecting the WMR within the context of the second experimental scenario is displayed. Similarly, Figure 16 illustrates the proposed control input applied to the wheeled mobile robot within the same context.

To quantitatively assess the tracking performance of the WMR, we employed the integral absolute error (IAE) and the control effort performance index as comparison metrics. The IAE is computed for each of the control strategies in the following manner:(75)$$\begin{array}{cc} {\mathrm{IAE}}_{i} & =\int_{0}^{t_{f}}\left|e_{i}\left(t\right)\right|dt.\;e_{i}\left(t\right)=\lambda_{i}\left(t\right)-\lambda_{id}\left(t\right), \end{array}$$
where $t_{f}$ is the total simulation duration and i=1,2, represents the position in the *x* and *y* direction, respectively. The control effort is given as follows:(76)$$\begin{array}{cc} P_{\mathrm{avg}} & =\frac{1}{N}\sum_{k=1}^{N}u^{2}\left(k\right) \end{array}$$
where *N* indicates the total count of samples. The associated key performance indicators IAE and $P_{\mathrm{avg}}$ for both strategies are provided in Table 1.

Examining the data in Table 1, it is evident that the FSADRC controller outperforms the BTC, FBTC, FADRC, and FSMC methods in terms of tracking performance. Although its tracking performance is nearly comparable to that of the BSADRC, the FSADRC requires minimal effort to accomplish its task compared to the BSADRC. This characteristic is particularly crucial in contexts where energy resources are limited, such as in mobile or autonomous applications. The enhanced efficiency of the FSADRC over the BSADRC is explained by the advantage of flatness control, which simplifies controller design by transforming the nonlinear system into a linear one. This feature makes all control based on the concept of flatness less complex than control based on backstepping. Ultimately, the experiment and table findings show that the disturbance rejection function simplifies the system model by addressing real-time modeling uncertainties. Consequently, the FSADRC method relies less on an exact analytical model description, treating unknown dynamics as internal disturbances compensated for by the rejection function. This enhances the robustness of FSADRC, which also incorporates the boundary layer technique to alleviate chattering effects.

## 7. Conclusions

This paper aims to introduce a robust control methodology for uncertain wheeled mobile robots (WMR). By employing flatness-based control, the nonlinear kinematic model of the WMR undergoes transformation into a canonical form, enabling the implementation of a robust feedback controller that incorporates boundary layer sliding mode control and extended state observer techniques. Simulation results conducted under various scenarios of uncertainties illustrate the effectiveness of FSADRC in enhancing the trajectory tracking performance of the WMR when compared to BTC, FBTC, and FADRC, even amid variations in slipping and external wind disturbances. Furthermore, within the same context, FSADRC demonstrates comparable efficiency to BSADRC in terms of trajectory tracking, while exhibiting an advantage in effort usage due to its flatness property. The smooth operation of FSADRC, coupled with its resilience against parameter variations and external disturbances, renders it a practical choice for real-world applications. Moreover, experimental findings using the TurtleBot3 validate the efficacy of the proposed FSADRC in real-world navigational tasks. In future studies, the application of FSADRC will extend to other robotic systems, such as quadrotors and arm manipulators, to assess its effectiveness and explore its potential for broader deployment.

## Figures and Tables

**Figure 1 sensors-24-02849-f001:**
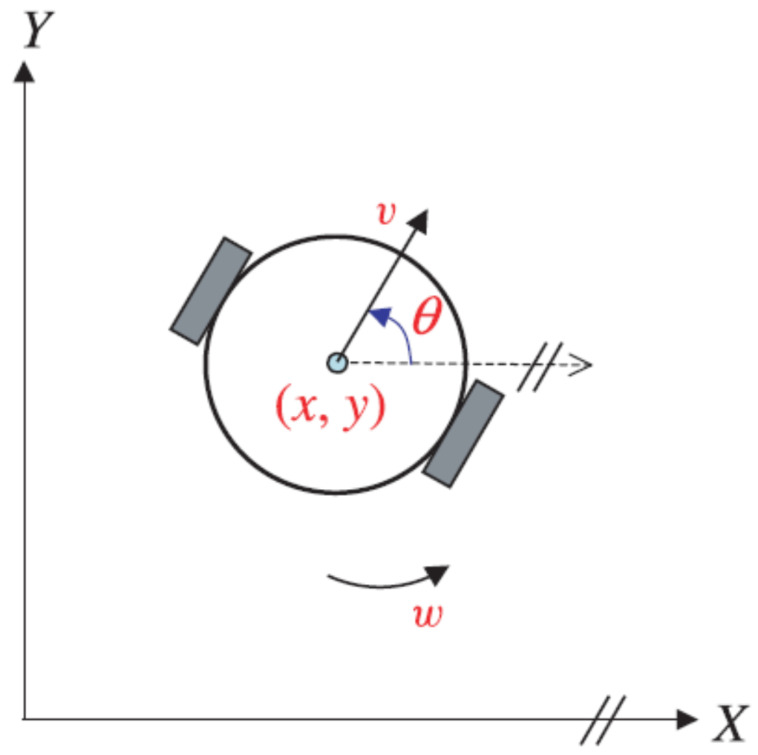
Two-wheeled mobile robot.

**Figure 2 sensors-24-02849-f002:**
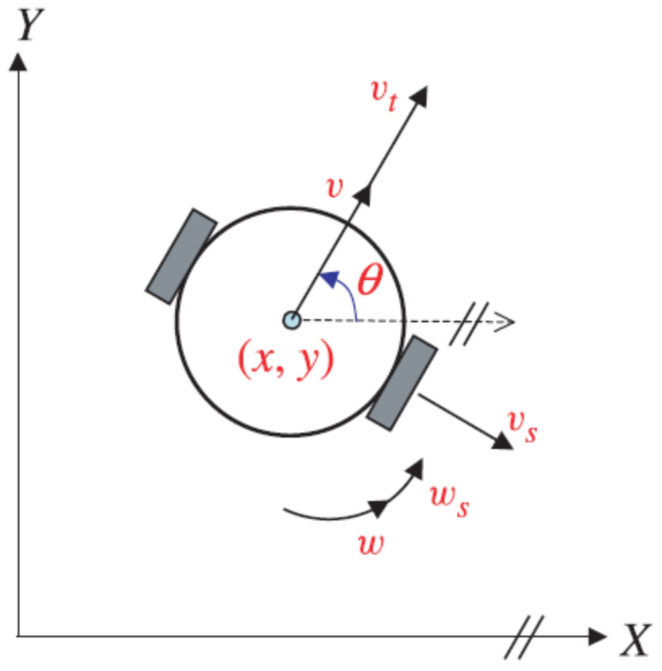
Two-wheeled mobile robot subject to uncertainties.

**Figure 3 sensors-24-02849-f003:**
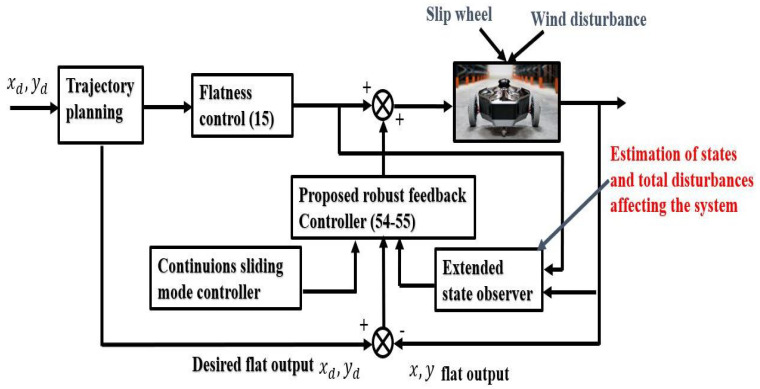
Mobile robot trajectory tracking control principle scheme.

**Figure 4 sensors-24-02849-f004:**
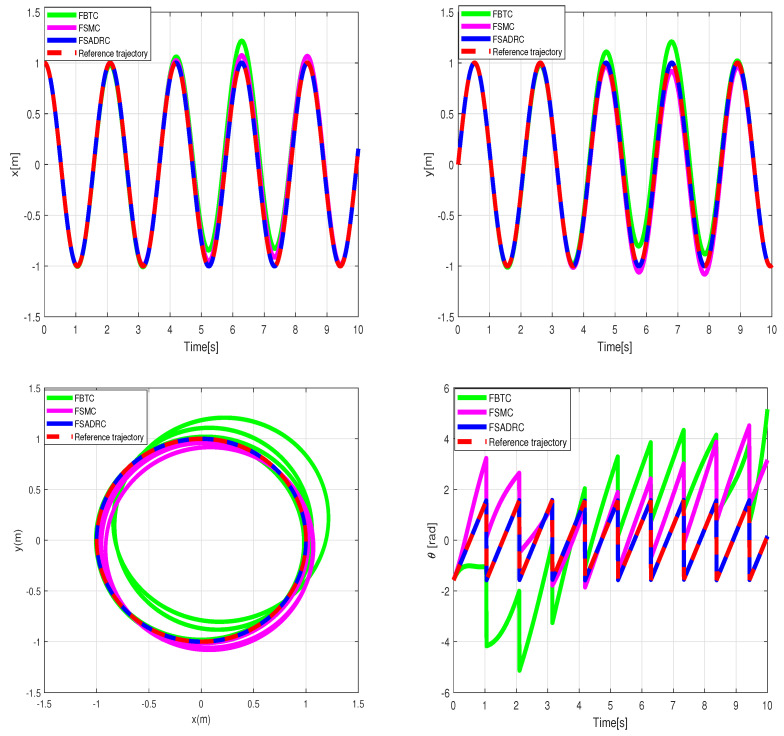
Simulation tracking results of the wheeled mobile robot under the conditions of the first scenario.

**Figure 5 sensors-24-02849-f005:**
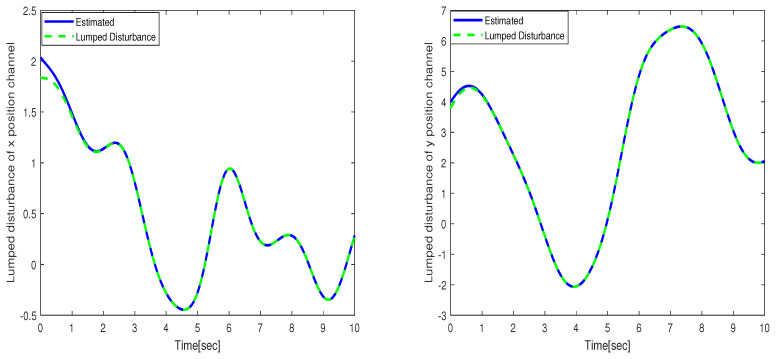
Lumped disturbance affecting the *x* and *y* position channels in the context of the first scenario.

**Figure 6 sensors-24-02849-f006:**
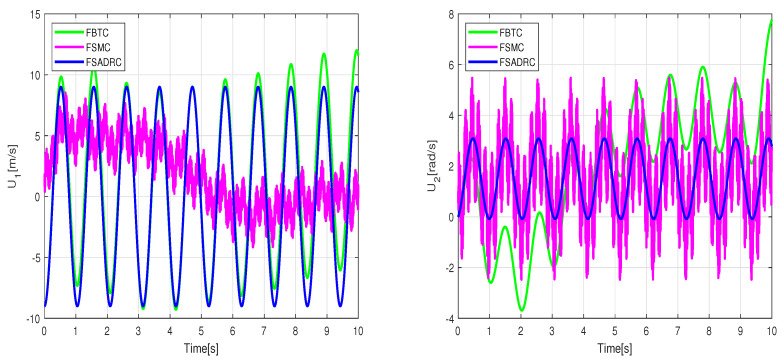
Control input applied to the wheeled mobile robot under the conditions of the first scenario.

**Figure 7 sensors-24-02849-f007:**
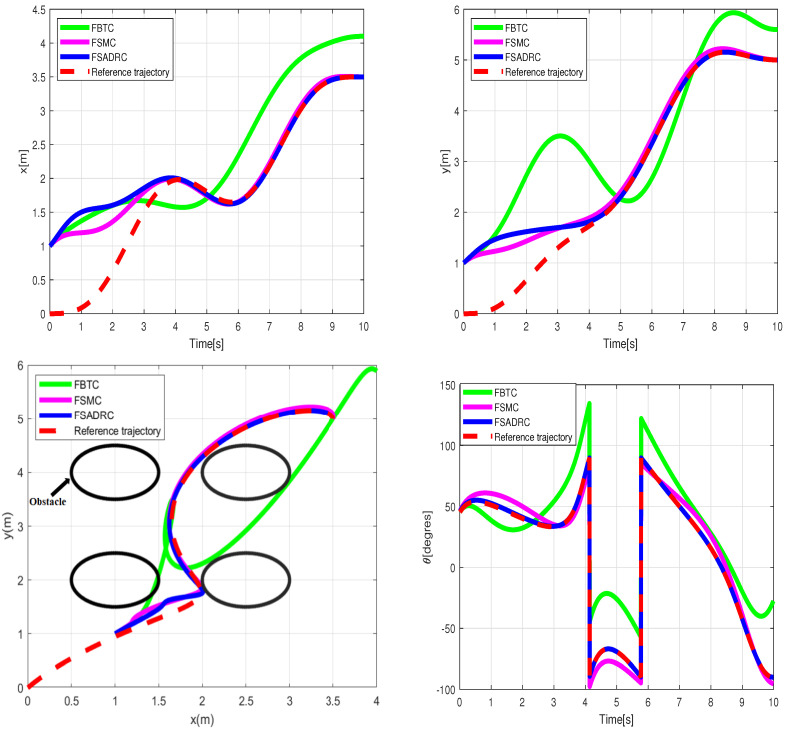
Simulation tracking results of the wheeled mobile robot in the conditions of the second scenario.

**Figure 8 sensors-24-02849-f008:**
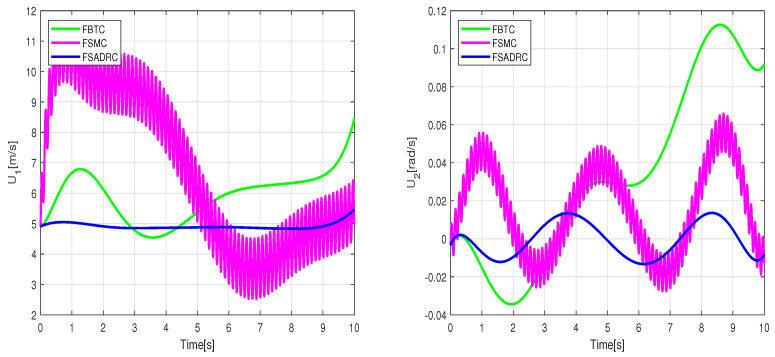
Control input applied to the wheeled mobile robot under the conditions of the second scenario.

**Figure 9 sensors-24-02849-f009:**
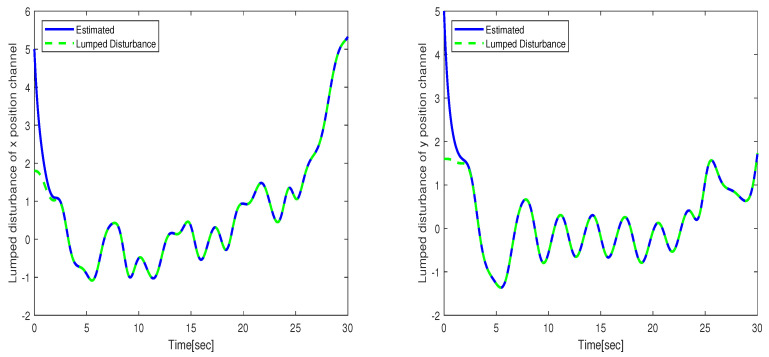
Lumped disturbance affecting the *x* and *y* position channels in the context of the second scenario.

**Figure 10 sensors-24-02849-f010:**
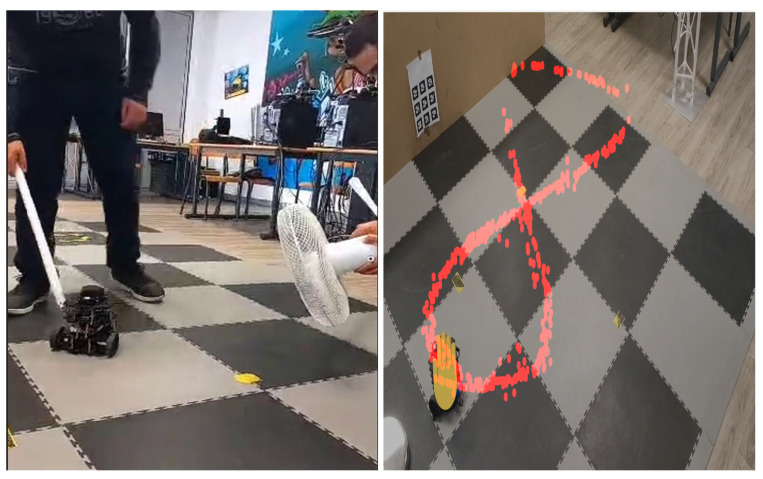
Real-time trajectory tracking experiment.

**Figure 11 sensors-24-02849-f011:**
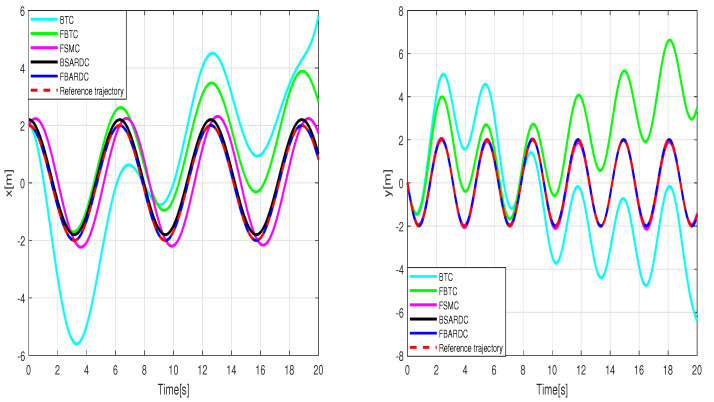
Results of the wheeled mobile robot’s tracking under the conditions of the first experiment scenario.

**Figure 12 sensors-24-02849-f012:**
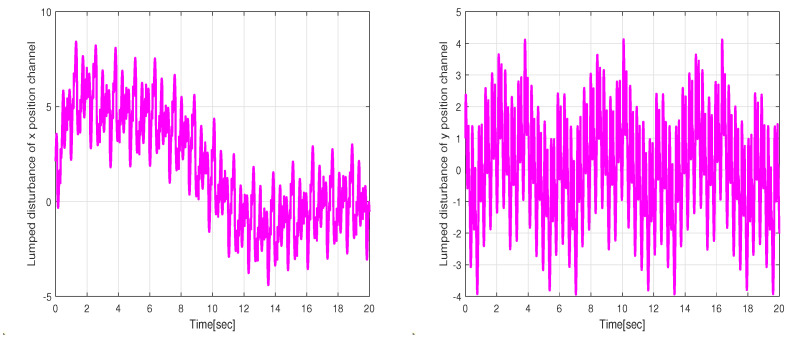
Estimation values of the lumped disturbances under the conditions of the first experiment scenario.

**Figure 13 sensors-24-02849-f013:**
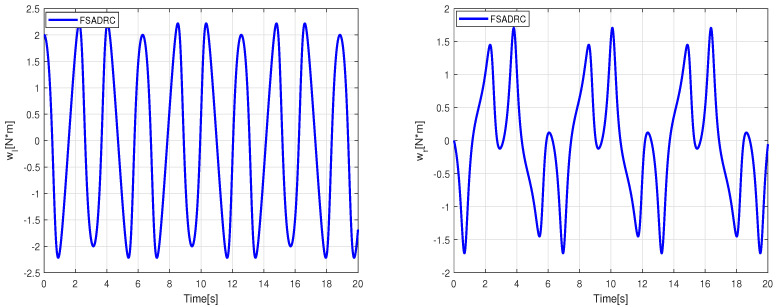
Control torques applied to the right and left wheels to track the eight-shaped reference trajectory.

**Figure 14 sensors-24-02849-f014:**
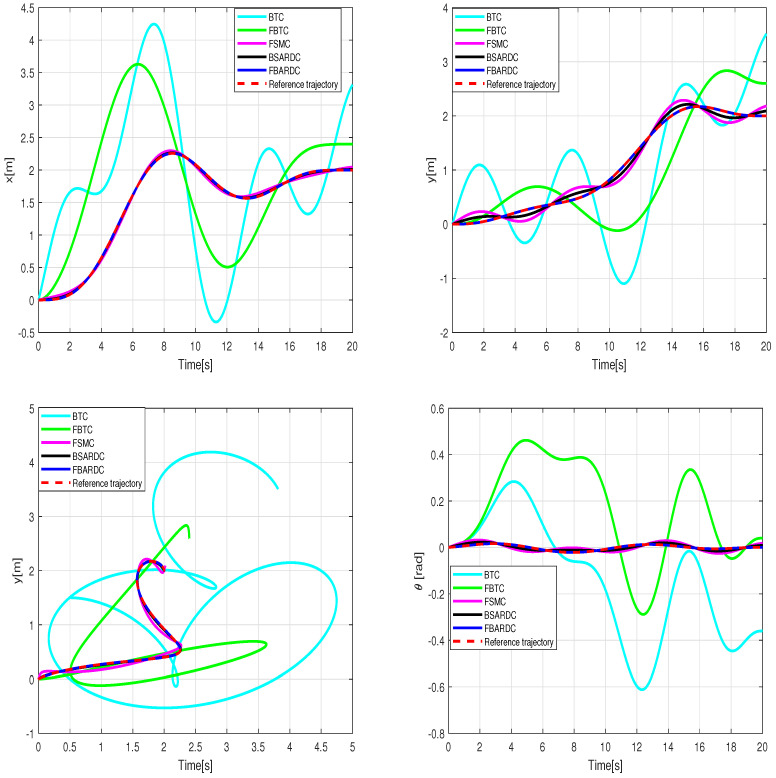
Results of the wheeled mobile robot’s tracking under the conditions of the second experiment scenario.

**Figure 15 sensors-24-02849-f015:**
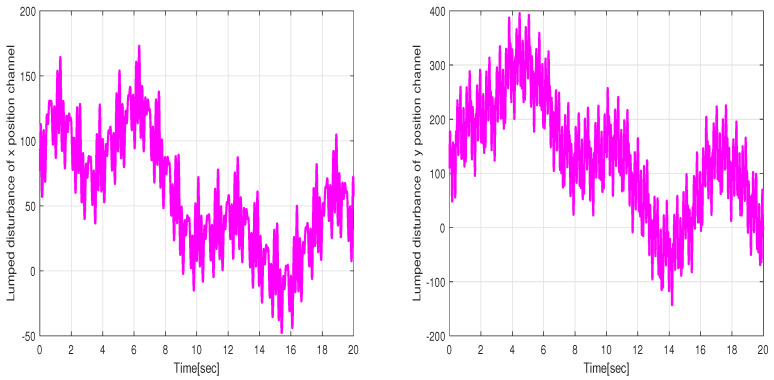
Estimated values of the lumped disturbances under the conditions of the second experiment scenario.

**Figure 16 sensors-24-02849-f016:**
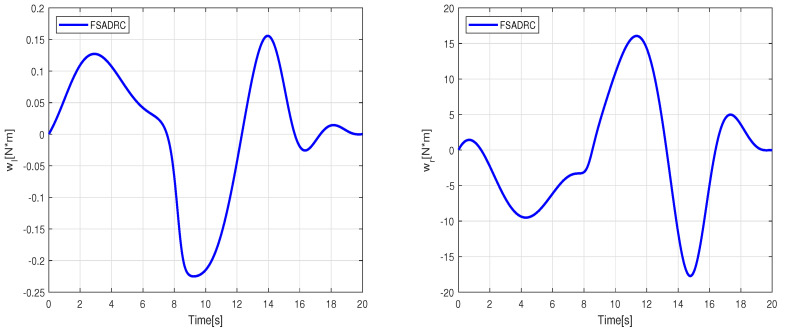
Control torques applied to the right and left wheels to track the Bézier reference trajectory.

**Table 1 sensors-24-02849-t001:** Performance indexes IAE and $P_{\mathrm{avg}}$.

Index	BTC	FBTC	FADRC	FSADRC	BSADRC
IAE	5.5351	4.2654	0.07	0.0127	0.02
$P_{\mathrm{avg}}$	2.5351	0.261	0.1266	0.13	1.253

## Data Availability

Data are contained within the article.

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
