# Peer review of "Robust Tracking Control of Wheeled Mobile Robot Based on Differential Flatness and Sliding Active Disturbance Rejection Control: Simulations and Experiments"

_sensors, 2024, doi:10.3390/s24092849_

Round 1
Reviewer 1 Report
Comments and Suggestions for Authors
In this paper, an ESO-based sliding mode control method is proposed to realize trajectory tracking of the wheeled mobile robot. Although slightly weaker in terms of innovation, the work is more complete and could be considered for publication after revision.
1. In the Introduction section, consider introducing comparisons among ESO, UDE, and EHGO to illustrate the advantages and necessity of ESO.
2. Equation (31) is wrong.
3. The stability proof is problematic and should be proved by combining the estimation error system with the tracking error system.
4. The saturated approximation of the sign function should be analyzed carefully in the proof process. The value of $a_s$ should be given in simulations and experiments
5. The formatting of references needs to be harmonized.
Comments on the Quality of English LanguageMinor editing of English language required.
Author Response
Thank you very much for taking the time to review this manuscript. Please find the detailed responses below and the corresponding revisions/corrections highlighted/in track changes in the re-submitted files

Reviewer 2 Report
Comments and Suggestions for Authors
After reviewing the provided paper "Robust Tracking Control of Wheeled Mobile Robot based on Differential Flatness and Sliding Active Disturbance Rejection control: Simulations and Experiments," I offer the following technical comments that could help in improving the paper further:
-
Clarification on Differential Flatness and its Application:
- The paper successfully introduces the concept of differential flatness and its application to the Wheeled Mobile Robot (WMR) system. However, it would be beneficial to include a more detailed explanation or a brief theoretical background on differential flatness for readers who may not be familiar with the concept. This could include its advantages over other linearization methods when applied to nonlinear systems like WMR.
-
Experimental Validation and Real-World Applicability:
- The experimental results section is comprehensive, showcasing the effectiveness of the proposed FSADRC controller. For further validation, consider comparing your proposed method with other state-of-the-art methods not only in simulation but also in real-world experiments to highlight its superiority in practical scenarios.
-
Robustness and Uncertainty Handling:
- The paper demonstrates robustness against uncertainties like slipping and wind disturbances. A deeper discussion on the types of uncertainties and disturbances the controller can handle would be beneficial. Additionally, quantifying the level of uncertainty the system can tolerate while maintaining performance metrics could enhance the paper's contribution to the field of robust control.
-
Chattering Phenomenon:
- It's commendable that the paper addresses the chattering issue commonly associated with Sliding Mode Control (SMC) through the integration of a boundary layer. A more detailed discussion on how the boundary layer thickness was selected and its impact on the system's dynamics and control performance would be valuable. This could include trade-offs between chattering reduction and tracking accuracy.
-
Comparison with Other Controllers:
- The comparative study is well-executed, showcasing the FSADRC's effectiveness over FBTC, FSMC, and other controllers. A more detailed statistical analysis or error metrics (such as RMS error) over multiple trials could provide a clearer picture of the performance improvements. Also, discussing the computational complexity of the proposed controller compared to others could provide insight into its feasibility for real-time applications.
NO comment for the author.
Author Response
Thank you very much for taking the time to review this manuscript. Please find the detailed responses below and the corresponding revisions/corrections highlighted/in track changes in the re-submitted files.

Reviewer 3 Report
Comments and Suggestions for Authors
This paper overall presents a theoretically solid design with a few simulations and experiments to prove its efficiency. However, I don't see much novelty in the proposed system. The core component of the system, ADRC, has been extensively studied as stated by the author themselves. And the paper does not show the significancy on adapting ADRC to the WMR control tracking domain. In addition, the trajectory in the experiment is too simplistic to showcase the performance gap between the proposed and baseline systems.
Comments on the Quality of English LanguageThis language in this paper is readable
Author Response

(The authors gave the same response as above.)

Round 2
Reviewer 1 Report
Comments and Suggestions for Authors
The current version of this paper can be published.